# Trajectory-BERT: Trajectory Estimation Based on BERT Trajectory Pre-Training Model and Particle Filter Algorithm

**DOI:** 10.3390/s23229120

**Published:** 2023-11-11

**Authors:** You Wu, Hongyi Yu, Jianping Du, Chenglong Ge

**Affiliations:** Information System Engineering College, PLA Strategic Support Force Information Engineering University, Zhengzhou 450001, China

**Keywords:** BERT, trajectory, particle filter, maximum posterior probability

## Abstract

In the realm of aviation, trajectory data play a crucial role in determining the target’s flight intentions and guaranteeing flight safety. However, the data collection process can be hindered by noise or signal interruptions, thus diminishing the precision of the data. This paper uses the bidirectional encoder representations from transformers (BERT) model to solve the problem by masking the high-precision automatic dependent survey broadcast (ADS-B) trajectory data and estimating the mask position value based on the front and rear trajectory points during BERT model training. Through this process, the model acquires knowledge of intricate motion patterns within the trajectory data and acquires the BERT pre-training Model. Afterwards, a refined particle filter algorithm is utilized to generate alternative trajectory sets for observation trajectory data that is prone to noise. Ultimately, the BERT trajectory pre-training model is supplied with the alternative trajectory set, and the optimal trajectory is determined by computing the maximum posterior probability. The results of the experiment show that the model has good performance and is stronger than traditional algorithms.

## 1. Introduction

In recent times, due to the advancement of high-precision positioning devices and target tracking technology [1], a great deal of trajectory data have been generated. Extensive research has been conducted in this direction in order to gain insight into target behavior, as trajectory data are the primary source of information [2]. By delving into the depths of data, people have made great strides in many areas, such as trajectory classification [3,4], prediction [5,6,7], tracking [8,9], and blind filling [10]. The aviation industry has seen a dramatic increase in air traffic, making it difficult for a single radar system to keep up with the changing regulations. Air traffic management and traffic control research [11,12,13] considers trajectory data to be an important reference point and therefore an in-depth analysis of trajectory data is essential.

In the context of receiving trajectory data, it is common for the quality of the received data is usually below standard, and there are a lot of noises and gaps in the data. The objective of this paper is to acquire the estimated trajectory that closely resembles the actual trajectory using the designed model, in cases where a trajectory with noise and missing points is acquired. Artificial modeling methods, such as Markov [14], Kalman filter [15], interacting multiple model [16] algorithm, etc., are used to pre-defined the motion model of the target in traditional methods. The trajectory data are then estimated and processed by calculating the maneuvering information such as velocity, acceleration, and steering rate in the data [17]. The validity of these features and the effectiveness of trajectory estimations will be influenced by the researchers’ experience. At the same time, due to the complexity of the target motion law, it is difficult to depict it accurately by artificial modeling.

The swift advancement of machine learning and deep learning has prompted numerous investigations into trajectory data mining. Xin Liu initially employed the technique in natural language processing (NLP) as a point of reference [18], treating trajectory points as “words”, trajectories as “sentences”, and constructing all trajectory points as “dictionaries”. The skip-gram model was employed to link the context and fill in the gaps in the trajectory points. Alessandro Crivellari employed the trajectory-to-vector approach [19] to transform the trajectory into a high-dimensional vector representation using Word2vec, and instructed the model to comprehend the correlation between trajectory points, thereby facilitating trajectory blind filling. Asif Nawaz employs a convolutional neural network to divide the map into grids [20], and utilizes a bidirectional convolutional recurrent encoder based on the attention mechanism to obtain trajectory feature information and anticipate the absent trajectory points. Alessandro Crivellari utilized the mask partial trajectory points [21] technique to train the BERT model, taking advantage of the Transformer’s impressive bidirectional representation capability, to anticipate and estimate the absent trajectory. Mashaal divided the region into hexagonal grids [22] and trained the BERT model to better capture the relationship between the six edges of each grid, so as to achieve the purpose of optimizing the missing trajectory.

Then, whether the above methods predict the trajectory by constructing a “dictionary” in the field of NLP, or predict the position after dividing the region into grids, they all belong to the training of discrete data. In the aviation field, aircraft motion has continuity, and treating aircraft motion as a discrete sequence obviously does not fit the realistic scenario. Based on the above problems, this paper proposes to use the BERT model to train the continuous trajectory sequence, and make the model learn the hidden motion law in the trajectory by masking part of the trajectory points. Then the particle filter algorithm is used to construct a trajectory candidate set for the observed noisy trajectory, and the candidate trajectory is sent to the model, which is determined by the maximum a posteriori criterion. The trajectory that best fits the true motion state is obtained.

The main contributions of this paper are as follows:The BERT model has been adapted to train sequences of continuous trajectory, and the pre-training model for the BERT trajectory has been acquired;For the task scenario, the BERT trajectory pre-training model and the particle filter algorithm are combined to construct a candidate trajectory set for the observed trajectory;The BERT trajectory pre-training model is used to calculate the maximum posterior probability of trajectory under trajectory sets and obtain the optimal trajectory estimation.

## 2. Methodology

### 2.1. The Task Scenario

In the actual trajectory signal receiving scenario, there will be errors in locating the target by existing means, as shown in Equation (1), where Y is the observed trajectory, X is the true trajectory, and W is the observation noise. To simplify the calculation, in this paper, the error W is treated as Gaussian noise with mean 0 and variance $\sigma^{2}$, as shown in Equation (2). The final application goal of this paper is to obtain an estimated trajectory X that is closest to the true trajectory through the BERT trajectory pre-training model based on an incomplete or noisy observed trajectory Y.
(1)Y=X+W
(2)$$W\sim N(0,\sigma^{2})$$

### 2.2. Decision Criteria

In trajectory estimation problems, the theoretical basis is the maximum a posteriori criterion, which maximizes the posterior probability by combining observations to calculate the likelihood probability. This criterion is widely used in the field of trajectory estimation, as shown in Equation (3).
(3)$$\max P(X|Y)=\frac{P(Y|X)\times P(X)}{P(Y)}$$

The maximum a posteriori probability is mainly determined by two parts: the likelihood probability P(Y|X) and the prior probability P(X). The denominator P(Y) can be regarded as a constant, which does not affect the decision result. The aim of this paper is to utilize deep learning to train the BERT model and express P(X). The clarity of the fact that Y represents the observed data is crucial, and when employing the BERT trajectory pre-training model to estimate the observed trajectory Y, X signifies the alternative trajectory set generated by Y through particle filter. $X_{1}$, $X_{2}$, ⋯, $X_{n}$ denotes a subset of the collection of alternative trajectories, as illustrated in Equation (4), where x denotes a point in the trajectory sequence and is a two-dimensional vector encompassing latitude and longitude, thus $X_{i}$ can be articulated as $X_{i}:\{x_{1},\;x_{2},\;\cdots ,\;x_{n}\}$, where $x_{i}:\{lon_{i},lat_{i}\}$. After the candidate trajectory set is obtained, the trajectory is fed into the BERT trajectory pre-training model by masking the trajectory point by point, and the corresponding prior probability $P(x_{i}|x_{\backslash i})$ is generated, where $x_{\backslash i}$ denotes all other points except point $x_{i}$. Extensive discussions have taken place in previous papers regarding the relationship between the prior joint distribution produced by BERT and the continuous multiplication of conditional probabilities [23,24,25], as illustrated in Equation (5). Once the BERT model is provided with an alternative trajectory and subjected to meticulous mask-by-mask validation, we can ultimately derive P(X) from the pre-training BERT trajectory model.
(4)$$\max P(X|Y)=\max\{P(X_{1}|Y),\;P(X_{2}|Y),\;\cdots ,\;P(X_{n}|Y)\}$$
(5)$$P(X)\approx P(x_{1}|x_{2}\cdots x_{i})\cdot P(x_{2}|x_{1},\;x_{3}\cdots x_{i})\cdots P(x_{i}|x_{1}\cdots x_{i-1})$$

In this task scenario, the estimated value is a two-dimensional (latitude and longitude) distribution problem, and the probability distribution of the estimated value of the model is assumed to conform to the Gaussian distribution. Therefore, the prior probability P(X) can be expressed in Equation (6), where Llon, Llat is the label, Plon and Plat is the estimated value of the model, and ∑ is the covariance matrix, as shown in Equation (7).
(6)$$P\left(X\right)=\frac{1}{{\left(2\pi \right)}^{\frac{2}{2}}|\sum {|}^{\frac{1}{2}}}\exp^{\left\{-\frac{1}{2}{\left(Llon-Plon,Llat-Plat\right)}^{T}\sum^{-1}(Llon-Plon,Llat-Plat)\right\}}$$
(7)$$\sum =\left(\begin{array}{cc} \sigma_{lonlon} & \sigma_{lonlat}\\ \sigma_{latlon} & \sigma_{latlat} \end{array}\right)$$

In summary, the procedure for calculating the maximum a posteriori probability is shown in Algorithm 1.
**Algorithm 1:** Procedure for calculating the maximum a posteriori probability1: Input an alternative trajectory $X_{1}$ and compute its prior probability $p(X_{1})$; 2: The input trajectory sequence is masked point by point and fed into the BERT trajectory pre-training model to calculate the prior probability of trajectory points: $p(x_{1}):Bert(mask,\;x_{2},\;x_{3},\;\cdots ,\;x_{n}),\;\cdots ,\;p(x_{n}):Bert(x_{1},\;x_{2},\;x_{3},\;\cdots ,\;mask)$; 3: The prior probability of the entire trajectory is estimated by multiplying the conditional probabilities one by one: $p(X_{1})\approx p(x_{1})\times p(x_{2})\times \cdots \times p(x_{n})$; 4: Calculate the Likelihood: 
$p(Y|X_{1})=\prod_{i}P(y_{i}|x_{i})=\sum_{i=1}^{n}\log P(y_{i}|x_{i})=\sum_{i=1}^{n}\log\frac{1}{\sqrt{2\pi \sigma^{2}}}\exp(-\frac{\left(y_{i}-x_{i}\right)}{2\sigma^{2}})$; 5: Calculate the posterior probability: $p(X_{1}|Y)=\frac{p(Y|X_{1})\times p(X_{1})}{p(Y)}$.

### 2.3. Trajectory Pre-Processing

In order to ensure that the BERT trajectory pre-training model achieves the desired training outcome, it is necessary to mask the trajectory data with a certain probability, as per the techniques employed in the realm of natural language processing. By hiding certain trajectory points in the input, specifically by substituting them with distinctive symbols, the model is able to anticipate the masked data by analyzing the data preceding and succeeding the mask. This training technique assists the model in understanding the relationship between the data, and by utilizing this approach, the BERT model can continuously modify weights to improve its learning efficiency and ability.

Using trajectory $X_{n}$ as a case in point, as depicted in Figure 1, where n signifies the trajectory’s length. To find the quantity of masks in this paper, consult paper [26] and apply a 15% probability to mask the trajectory sequence. In contrast with other discrete sequences, the model is a continuously sequenced process, making it impossible to use a special symbol to represent the position of the mask. In this article, the mask point is set to −9999 for the task scenario in this paper, and to comply with the unified requirements for sequence length when calculating the BERT model, the position where padding must be 0 is specified. The BERT model is expected to be able to learn the meanings of mask and padding represented by −9999 and 0 respectively through training. The purpose of training is to learn how to estimate missing positions based on contextual information by processing the input trajectory with the mask and producing the corresponding mask position as the output.

## 3. Train Model

### 3.1. Model Structure

The encoder module of the transformer, or BERT model, as illustrated in Figure 2, is the model employed in this situation. It is important to remember that when it comes to natural language processing, the BERT model only processes discrete data. The output value is the maximum probability value in the dictionary when predicting the output, which can be classified as a classification problem. This chapter’s data are continuous, and the output is an estimated value. Consequently, to achieve the goal of trajectory estimation, this chapter modifies the module to better suit the task scenario.

Referring to the mask language model (MLM) method [26] when using BERT to train language models, this paper proposes a mask trajectory model (MTM) training method suitable for this application scenario, as shown in Figure 3. The mask trajectory data are fed into the model. Thus, the model can predict the missing trajectory through the front and back trajectory. Simultaneously, in order to enhance the training effect of the model, we add a fully connected layer on the basis of MTM in parallel. The fully connected layer is similar to the traditional encoder-decoder method, which realizes the function of mapping low-dimensional features to high-dimensional and then restoring low-dimensional features from high-dimensional to low-dimensional, so as to better extract data features. MTM is to restore the position of the mask, that is, the local trajectory, which has certain limitations, while the parallel FC-decoder layer can extract the feature of the trajectory from a global perspective. The combination of the two makes the model training better, which is verified in the subsequent experimental link.

In the expected goal, the output of the BERT model set in this paper is the estimated latitude and longitude value. Compared with Equation (6), it can be found that the model cannot express the covariance matrix, so the output of the model needs to be further modified. The solution of this paper is to send the output of FC-decoder into a one-layer fully connected layer, as shown in Figure 4. The 2-dimensional output is extended to 6-dimensional output, which is used to generate the prediction value and covariance matrix.

### 3.2. Loss Function

Considering that the function of the model is a prediction problem rather than a classification problem, this paper uses the SmoothL1Loss (SML1Loss) commonly used in prediction tasks, as shown in Equation (8).
(8)$$SML1Loss=\left\{\begin{array}{l} 0.5{\left(x_{i}-x_{i}^{p}\right)}^{2},if|x_{i}-x_{i}^{p}|<1\\ |x_{i}-x_{i}^{p}|-0.5,\mathrm{otherwise} \end{array}\right.$$
where $x_{i}$ is the true value, $x_{i}^{p}$ is the predicted value, when $\left|x_{i}-x_{i}^{p}\right|$ is less than 1, the square error is used, otherwise the linear error is used, which makes the SML1Loss can punish outliers (such as too large or too small outliers) less than the mean square error (MSE) and mean absolute error (MAE) loss function, so that the model is more robust. To avoid zero non-differentiable and gradient explosion, the training effects of the above three loss functions are compared in the subsequent chapters. The overall loss of the model is divided into three parts: covariance matrix (Loss1), decoder (Loss2), and MTM (Loss3), as shown in Equation (9).
(9)Loss=Loss1+Loss2+Loss3

Among these three loss functions, Loss1 is the loss function of the covariance matrix generated by the model, which needs to be constructed by ourselves. Here, this paper proposes two solutions. The first one is shown in Equation (10).
(10)Loss1.1=SmL1Loss(cov−seq_cov)
where cov is the covariance generated by the model and seq_cov is the covariance of the training data; The second method is given in Equation (11).
(11)Loss1.2=SML1oss(cov−seq_cov)+SML1oss(decoder_cov−seq_cov)
where decoder_cov is the covariance generated by the FC-decoder layer. It can be seen from the formula that the first loss function is more targeted, while the second loss is more restrictive from a global perspective, but the expected convergence effect is not as good as the first loss function. In this paper, for these two different loss functions, comparative experiments are carried out under the same conditions in the subsequent experimental sections.

## 4. Set of Alternative Trajectories

### 4.1. Particle Filter

When using the particle filter algorithm, the state equation and the observation equation should be clarified first, as shown in Equations (12) and (13), where f(x) is called the state transition function, h(x) is called the observation function, $Q_{k}$ is the process noise, and $R_{k}$ is the observation noise, $Q_{k}$ and $R_{k}$ are treated as Gaussian noise with mean 0 and variance $\sigma^{2}$.
(12)$$X_{k}=f(X_{k-1})+Q_{k}$$
(13)$$Y_{k}=h(X_{k})+R_{k}$$

The basic concept of particle filter is to depict the posterior distribution of states utilizing a set of random samples with corresponding weights (commonly referred to as “particles”). The posterior distribution in this application scenario pertains to the distribution of every trajectory point along the observation trajectory. The weights and positions of the particles are adjusted based on the state observations after obtaining some random samples with corresponding weights. These samples are then used to approximate the posterior distribution of the state. Finally, the state is estimated by considering the weighted sum of these samples. Particle filters are not limited by the linear and Gaussian assumptions of the system model and describe the probability density of state variables in a sample rather than a function. This means that it does not need too many constraints on the probability distribution of the state variables, so it is widely used in nonlinear and non-Gaussian dynamic systems.

This paper postulates that the probability distribution of the observed trajectory points follows a Gaussian distribution, allowing for the representation of the probability density of the trajectory points by employing a particle filter to approximate the probability density. Over the course of this duration, the particles generated can be regarded as potential points of trajectory that were formed in close proximity to the observed points of trajectory. Through the consolidation and amalgamation of multiple potential trajectory points, a collection of trajectory candidates is generated, and the real trajectory is incorporated into the collection of trajectory candidates. Algorithm 2 provides a comprehensive outline of the entire process for the traditional particle filter.
**Algorithm 2:** Process of traditional particle filter algorithm1: Setting initial values: $X_{0}\sim N(\mu ,\sigma^{2})$; 2: Generate particles and initial weights, and set the number of generated particles as n: $X_{0}^{\left(i\right)},W_{0}^{\left(i\right)}=\frac{1}{n}$; 3: Prediction step: $X_{1}^{\left(i\right)}=f(X_{0}^{\left(i\right)})+V$, V~N(0,Q); 4: Update step: the observation is assumed to be $y_{1}$, $W_{1}^{\left(i\right)}=f_{R}(y_{1}-h(x_{1}^{\left(i\right)}))\cdot W_{0}^{\left(i\right)}$,   $f_{R}$ is the probability density function of the observation noise; 5: Weights are normalized: $W_{1}^{\left(i\right)}=\frac{W_{1}^{\left(i\right)}}{\sum W_{1}^{\left(i\right)}}$; 6: The new particle $X_{1}^{\left(i\right)}$ and the new weight $W_{1}^{\left(i\right)}$ are calculated; 7: Repeat steps 3, 4, and 5 to obtain new particles and weights in a cycle

In the follow-up work of this paper, the equation of state in traditional particle filter algorithm is not involved. The purpose of training the BERT trajectory pre-training model in this paper is to learn the change law between trajectory points in the way of deep learning. Therefore, after the initial n particles are obtained, these particles are sent to the BERT trajectory pre-training model respectively, and the BERT trajectory pre-training model is used to replace the state transition equation f(x) to generate the particles at the next moment, so as to complete the work of predicting and updating particles.

If m points are collected when sampling the probability density of each observation point and the length of the observed trajectory is n, then there are $m^{n}$ alternative sets of trajectories, as shown in Figure 5 for simple effect. It can be inferred that when there are enough sampling points, the real trajectory will be included. The goal of this paper is to determine which trajectory is the closest to the real trajectory through the BERT trajectory pre-training model.

### 4.2. Construct the Set of Alternative Trajectories

In this paper, BERT pre-training model is used to replace the state transition equation in the traditional particle filter algorithm. In order to explore what method is used to generate the best particle effect, this paper makes two improvements to the traditional particle algorithm, among which Algorithm A is shown in Algorithm 3.
**Algorithm 3:** Improved particle filter Algorithm A**Trajectory of observation:** $Y:\{y_{1},\;y_{2},\;\cdots ,\;y_{n}\}$; 1: Create the primary particles: $x_{\left(1,m\right)}=y_{1}+\mathrm{random}(m)$; 2: Obtain *m* trajectories with varying initial observations: $Y_{1}:\{x_{\left(1,m\right)}^{1},\;y_{2},\;\cdots ,\;y_{n}\},\;Y_{2}:\{x_{\left(1,m\right)}^{2},\;y_{2},\;\cdots ,\;y_{n}\},\;\cdots ,\;Y_{m}:\{x_{\left(1,m\right)}^{m},\;y_{2},\;\cdots ,\;y_{n}\}$; 3: Mask the trajectory data: $Y_{1}:\{x_{\left(1,m\right)}^{1},\;mask,\;\cdots ,\;y_{n}\}$; 4: The second batch of particles is generated by the BERT trajectory pre-training model: $Y_{1}:\{x_{\left(1,m\right)}^{1},\;x_{\left(2,m\right)}^{1},\;\cdots ,\;y_{n}\}$; 5: Mask the trajectory data: $Y_{1}:\{y_{1},\;x_{\left(2,m\right)}^{1},\;mask,\;\cdots ,\;y_{n}\}$; 6: The third batch of particles is generated by the BERT trajectory pre-training model: $Y_{1}:\{y_{1},\;x_{\left(2,m\right)}^{1},\;x_{\left(3,m\right)}^{1},\;\cdots ,\;y_{n}\}$; 7: Alternate trajectories are obtained by generating particles one by one in order: $X_{1}:\{x_{\left(1,m\right)}^{1},\;x_{\left(2,m\right)}^{1},\;\cdots ,\;x_{\left(n,m\right)}^{1}\}$.

In Algorithm A, the first batch of particles is obtained by adding random noise (Gaussian distribution with mean 0 and variance 0.3) to the first trajectory point of observation trajectory data, where m represents the number of generated particles, and the first observation point is replaced by m particles, respectively, so that different trajectories of m initial observation points are obtained, where $x_{\left(1,m\right)}^{i}(i=1,\;2,\;\cdots ,\;m)$ represents the ith particle in the first batch of m particles, and $Y_{i}(i=1,\;2,\;\cdots ,\;m)$ represents the ith trajectory constructed. Taking the first trajectory $Y_{1}$ as an example, in order to generate the next batch of particles, the observation trajectory point after the first batch of particles is masked and sent to the BERT trajectory pre-training model, and the prediction data generated by the model is the second batch of particles. In the subsequent process, the above operations are repeated to obtain all particles in order. It should be noted here that when generating the particle at the next time, only the particles at the previous time are used, and the rest of the data in the sequence are still the observation data. Algorithm A is adjusted to obtain Algorithm B as shown in Algorithm 4.

**Algorithm 4:** Improved particle filter Algorithm B**Trajectory of Observation:** $Y:\{y_{1},\;y_{2},\;\cdots ,\;y_{n}\}$; 1: Create the primary particles: $x_{\left(1,m\right)}=y_{1}+\mathrm{random}(m)$; 2: Obtain *m* trajectories with varying initial observations: $Y_{1}:\{x_{\left(1,m\right)}^{1},\;y_{2},\;\cdots ,\;y_{n}\},Y_{2}:\{x_{\left(1,m\right)}^{2},\;y_{2},\;\cdots ,\;y_{n}\},\;\cdots ,\;Y_{m}:\{x_{\left(1,m\right)}^{m},\;y_{2},\;\cdots ,\;y_{n}\}$; 3: Mask the trajectory data: $Y_{1}:\{x_{\left(1,m\right)}^{1},\;mask,\;\cdots ,\;y_{n}\}$; 4: The second batch of particles is generated by the BERT trajectory pre-training model: $Y_{1}:\{x_{\left(1,m\right)}^{1},\;x_{\left(2,m\right)}^{1},\;\cdots ,\;y_{n}\}$; 5: Mask the trajectory data: $Y_{1}:\{x_{\left(1,m\right)}^{1},\;x_{\left(2,m\right)}^{1},\;mask,\;\cdots ,\;y_{n}\}$; 6: The third batch of particles is generated by the BERT trajectory pre-training model: $Y_{1}:\{x_{\left(1,m\right)}^{1},x_{\left(2,m\right)}^{1},\;x_{\left(3,m\right)}^{1},\;\cdots ,\;y_{n}\}$; 7: Alternate trajectories are obtained by generating particles one by one in order: $X_{1}:\{x_{\left(1,m\right)}^{1},\;x_{\left(2,m\right)}^{1},\;\cdots ,\;x_{\left(n,m\right)}^{1}\}$.

In Algorithm B, the first and second particles are generated in the same method as in Algorithm A, but the third batch is generated in a different way, with the resulting particles constantly replacing the observed data. Therefore, when predicting the particles the next time, all the generated particles before the current time and the observed data after the current time are involved in the prediction. Figure 6 shows the process of constructing the set of alternative trajectories using the particle filter algorithm. After all the particles are obtained, the Cartesian product algorithm is used to cross-arrange and combine the particles to obtain the alternative trajectory set.

## 5. Experimental Analysis

### 5.1. Evaluation Indicators

Considering that the problem studied in this paper is trajectory estimation, the mean absolute error (MAE), which is widely used in this field, is used as an evaluation index to measure the quality of trajectory prediction. At the same time, in order to compare the intuitive difference between the estimated trajectory and the real trajectory, this paper defines the average distance deviation (MEDV) to more intuitively show the accuracy of the model, as shown in Equations (14) and (15).
(14)$$MAE=\frac{1}{n}\sum_{i=1}^{n}(|A_{i,Lat}-E_{i,Lat}|+|A_{i,Lon}-E_{i,Lon}|)$$
(15)$$MEDV=\frac{1}{n}\sum_{1}^{n}distance(A_{i},E_{i})$$
where n is the number of samples, $A_{i}$ is the predicted trajectory, $E_{i}$ is the actual trajectory, and distance is the algorithm used to calculate the geographical distance between different latitude and longitude. The smaller the values of these two evaluation indexes are, the closer the estimated trajectory is to the actual trajectory, and the estimation effect is better.

### 5.2. Datasets

Based on the development of deep learning, neural networks can model complex motion laws, and can estimate trajectory more accurately than traditional methods. In order to train the trajectory pre-training model with better performance and enable the model to learn the complex motion laws contained in the trajectory, it is necessary to select high-precision trajectory data. The data used in this paper are from the public ADS-B data set from January to May 2021 provided by the OpenSky website, which meets the requirements of the model for data accuracy. By analyzing this dataset, it is found that there are problems of duplication and missing trajectory points, so it is necessary to clean and reconstruct this dataset first. Related work mainly includes trajectory selection in specific regions, outlier cleaning, trajectory segmentation, linear interpolation, data normalization, sliding window data segmentation, etc. Paper [27] has made a detailed explanation, and is not repeated here.

The experimental data use the L2J model data in the real ADS-B data, which is a civil aircraft. The data scale is large, the trajectory is relatively complete, and the maneuvering law is diverse, which can fully support the training of the model. After cleaning and reconstruction, a total of 3600 trajectory data were obtained, and 300,000 training data were obtained after sliding window partition, of which 65% of the sample trajectory data were used as the training set, 25% of the data were used as the validation set, and 10% of the data were used as the test set. In the construction of observation data, Gaussian noise with mean 0 and variance 0.3 is added to the real ADS-B in this paper.

### 5.3. Experimental Parameter Settings

The Adam optimizer is used in model training, weight decay is set to l^−5^, learning rate is l^−3^, training batch size is set to 1024, and a graphics card with 24 G video memory is used for training, the detailed parameters of the experiment are shown in Table 1.

### 5.4. Analysis of Experimental Results

#### 5.4.1. Comparison of Different Loss Functions

Figure 7 compares the training effects of three loss functions. It can be seen that the variable loss function SML1Loss has the best effect, which is more suitable for the task scenario of this chapter. At the same time, in order to verify the performance of calculating covariance loss function, a comparison experiment is carried out under the condition of using SML1Loss loss function. The experimental results are shown in Figure 8. It is easy to see that the training effect of simple and direct calculation of covariance loss function is better, and the learning of the model for covariance is more focused on the data at the position of the mask. In the subsequent experiments, the above two loss functions are used.

#### 5.4.2. Functional Analysis of the FC-Decode Layer

In this subsection, in order to verify the effect of the FC-decoder layer on BERT model training, the comparison experiment of removing the FC-decoder layer and only retaining the MTM layer is performed. The results are shown in Figure 9. It can be seen that MTM combined with FC-decoder can effectively improve the training effect. The analysis is because FC-decoder can focus on the characteristics of the trajectory at the global level and pay more attention to the global characteristics of the trajectory, while MTM focuses on restoring the information of the masked position and pays more attention to the local characteristics of the trajectory. The combination of the two can extract trajectory features more effectively, and improve the performance of the model.

#### 5.4.3. Effect Analysis of Positional Embedding and Token Embedding

In order to verify the role of positional embedding and token embedding, the following comparative experiment is performed, as shown in Figure 10. It can be seen that the use of token embedding plays a great positive role in model training. For deep learning, a small dimension means insufficient description of the inherent laws of data, while dimension expansion after token embedding can better describe the characteristics of data [28]. In addition, there is time correlation between trajectory points, and the time characteristics cannot be well captured due to parallel calculation during model training. Therefore, by adding positional embedding, the model can learn the location information between data to improve the training performance of the model.

#### 5.4.4. Determine the Optimal Parameters of the Model

After determining the loss function and structure of the model, this paper conducts a combination exploration of the input dimension, output dimension, the number of heads and the number of encoder layers of the model, and makes the following eight sets of comparison experiments to find the optimal model parameters, as shown in Table 2. The experimental results are shown in Figure 11. As can be seen, parameter 6 is the best.

#### 5.4.5. Verify the Model Training Effect

The BERT trajectory pre-training model proposed in this paper is trained on 15% of the trajectory points in the mask sequence, so that the model can predict the value of the mask position from the rest of the trajectory points. In order to verify whether the model has learned the inherent movement law of the data during the training, the verification experiment in this section selects a test track and removes part of the data according to the proportion of 20%, 40% and 80%, to simulate the situation that the receiving track is interrupted due to signal interruption and other factors in the actual signal receiving scene. It needs to be explained that this is only to verify whether the model training is effective. In the verification, the incomplete trajectory is fed into the BERT trajectory pre-training model to see whether the model can complete the missing points through the incomplete trajectory. The experimental results are shown in Figure 12.

The experimental results show that the model can fill the missing points through the incomplete trajectory, and even in the case of the trajectory missing 80% of points, it can still obtain more accurate estimation results through the front and back trajectories, and restore the trajectory in line with the real movement trend. Therefore, it can be determined that the training of the model is effective. It is proved that the model can learn the movement law of the data by masking the trajectory points. In this section, MAE and MEDV are used as evaluation indicators to show the performance of the model more intuitively, as shown in Table 3.

#### 5.4.6. Comparison of Performance of Two Particle Filter Algorithms

According to the above experiments, the optimal parameters of the model are determined and the model is trained according to these parameters, and the BERT trajectory pre-training model is obtained. The effectiveness of model training is verified by experiments. In order to compare the performance of the two particle filter algorithms, this section makes an experimental comparison between the two algorithms and without particle filter candidates. In the experiment, a lot of memory space will be occupied when constructing the candidate trajectory set. Limited by the computer memory, 500,000 candidate trajectories are finally constructed with the particle number of 15, and the candidate trajectories are sent to the model and the maximum a posteriori is calculated. Finally, the trajectory with the largest posterior probability value is selected as the final estimated trajectory, and the experimental results are shown in Figure 13. It can be seen that the performance of Algorithm A is better than that of Algorithm B. In Algorithm B, the analysis is due to the fact that when predicting the particle at the next time, the model cannot make good use of the observation data to correct the particle, which leads to the accumulation of errors and the performance degradation. The observed trajectory also belongs to the candidate trajectory set in essence, and because it is more consistent with the movement trend of the real trajectory, it will also show good performance when using BERT alone. However, as the number of particles increases, we can infer that the performance of BERT combined with particle filtering will be further improved. In order to visually compare the performance differences, MAE and MEDV are selected as the evaluation indicators to measure the performance of the algorithm in the experiment in this section, as shown in Table 4.

#### 5.4.7. Compare Kalman Filter Algorithms

In order to evaluate the performance of the BERT trajectory pre-training model, this section makes a comparison experiment with the Kalman filter method which is widely used in the field of trajectory optimization. The key idea of the Kalman filter is to combine the state equation with the observation equation, obtain the optimal solution, use the final optimal solution to predict the current value, and modify the current value and the current covariance matrix with the observation value. The observed value is understood as the co-ordinates obtained in practice, and there is observation error; the predicted value is the motion law obtained by the mathematical model, and there is a situation inconsistent with the actual situation. It uses the covariance matrix of the state estimation error to measure the reliability of the estimation (Kalman gain). If the predicted state is reliable, the Kalman gain is small; if the observed state is reliable, the Kalman gain is large, and the estimated value and covariance matrix are updated in an optimized way. The Kalman filter algorithm is shown in Algorithm 5.
**Algorithm 5:** Kalman filter algorithm process1: A priori estimate: $x_{k}^{-}=A\cdot x_{k-1}^{+}+B\cdot \mu_{k-1}+Q_{k}$
2: The covariance of the prior estimate: $P_{k}^{-}=AP_{k-1}^{+}A^{T}+Q_{k}$
3: Calculate Kalman gain: $K_{k}=\frac{H^{T}P_{k}^{-}}{HP_{k}^{-}H^{T}+R_{k}}$
4: Update the estimate: $x_{k}^{+}=K_{k}(y_{k}-Hx_{k}^{-})+x_{k}^{-}$
5: Update the covariance matrix: $P_{k}^{+}=(1-K_{k}H)P_{k}^{-}$

When using the Kalman filter algorithm, the state equation in line with the motion scene should be constructed first. Considering that only latitude and longitude are used as input data when training the BERT trajectory pre-training model, the Kalman filter algorithm with constant velocity is selected as a comparison for experimental verification. Its state equation and observation equation are as follows:(16)$$\left[\begin{array}{l} x_{k+1}\\ y_{k+1}\\ v_{x,k+1}\\ v_{y,k+1} \end{array}\right]=\left[\begin{array}{llll} 1 & 0 & T & 0\\ 0 & 1 & 0 & T\\ 0 & 0 & 1 & 0\\ 0 & 0 & 0 & 1 \end{array}\right]\left[\begin{array}{l} x_{k}\\ y_{k}\\ v_{x,k}\\ v_{y,k} \end{array}\right]+\left[\begin{array}{ll} 0.5a_{x}T^{2} & 0\\ 0 & 0.5a_{y}T^{2}\\ a_{x}T & 0\\ 0 & a_{y}T \end{array}\right]+Q_{k}$$
(17)$$z_{k}=\left[\begin{array}{llll} 1 & 0 & 0 & 0\\ 0 & 1 & 0 & 0 \end{array}\right]\cdot \left[\begin{array}{l} x_{k+1}\\ y_{k+1}\\ v_{x,k+1}\\ v_{y,k+1} \end{array}\right]+R_{k}$$

By comparing the experimental results shown in Figure 14, it can be seen that the optimized trajectory of the Kalman filter is more dependent on the observed data, that is, the optimization is carried out on the basis of the observed data, and is also affected by the equation of motion. The state equation is used to express the law of motion, which is equivalent to adding constraints to the motion and restricting the regional scope when locating the next point. Although this method can lead to more accurate positioning accuracy, there are still many errors when describing the motion state of the aircraft. The BERT trajectory pre-training model is obtained by training a large number of historical trajectories. Its internal parameters have been able to better express the trajectory motion law when dealing with noisy observation data, so the real trajectory can be smoothly and accurately optimized from the observation data, meaning the performance is more prominent, as shown in Table 5.

Because the BERT pre-training model uses a variety of data during training, it can also perform well in the face of the trajectory of different sports types. Kalman filters do not work well when trajectories are in complex and variable motion patterns, such as heading mutations, as shown in Figure 15. Their generalization performance is poor due to their heavy reliance on observations and state equations, while the BERT pre-training models can still optimize trajectories well, demonstrating the validity of the methods presented in this chapter, as shown in Table 6.

## 6. Conclusions

Aiming at a key problem—the difficulty of manually modeling complex trajectories in real scenarios—this paper proposes a trajectory-estimation method based on a BERT trajectory pre-training model combined with a particle filter algorithm. Firstly, the real ADS-B data were cleaned and reconstructed, and then sent to the BERT model with adjusted structure for training after mask processing. Then, it explored the optimal parameters of the model and verified the training effect of the model through a series of verification experiments. Then, the particle filter algorithm was used to construct the trajectory set to verify the performance of the model. Finally, the comparison experiment with the traditional Kalman filter algorithm shows the advantages of the BERT trajectory pre-training model. However, due to the limitation of hardware conditions, it is impossible to traverse all possible candidate trajectories. In theory, when the number of particles is enough and the set of candidate trajectories is complete enough, the true trajectory can be restored by calculating the maximum a posteriori. The experiments in this paper prove the feasibility of the method, which provides a new research idea for the problem of trajectory estimation.

## Figures and Tables

**Figure 1 sensors-23-09120-f001:**
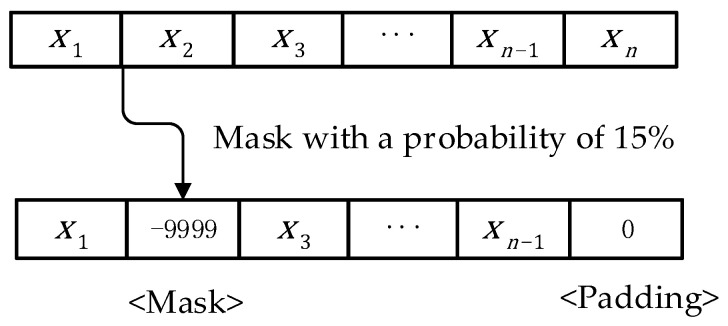
Trajectory preprocessing.

**Figure 2 sensors-23-09120-f002:**
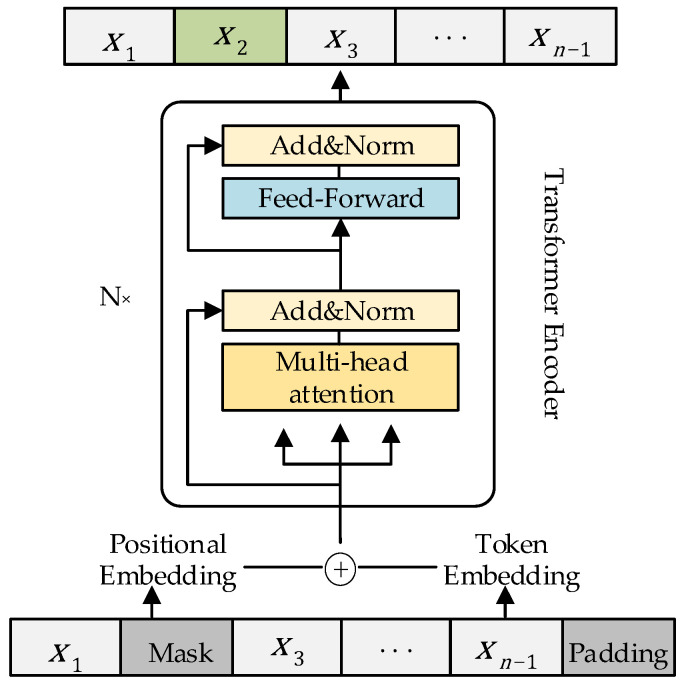
BERT Model Architecture.

**Figure 3 sensors-23-09120-f003:**
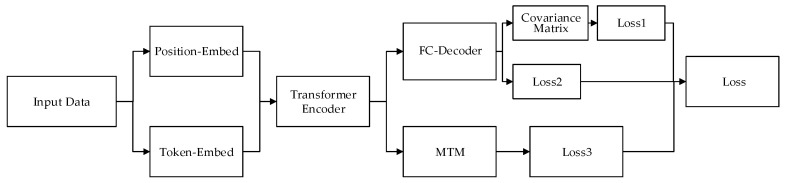
Structure of BERT trajectory pre-training model.

**Figure 4 sensors-23-09120-f004:**
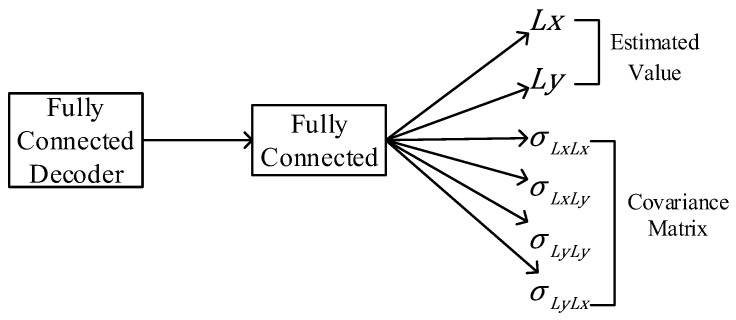
Schematic of the covariance matrix generation.

**Figure 5 sensors-23-09120-f005:**
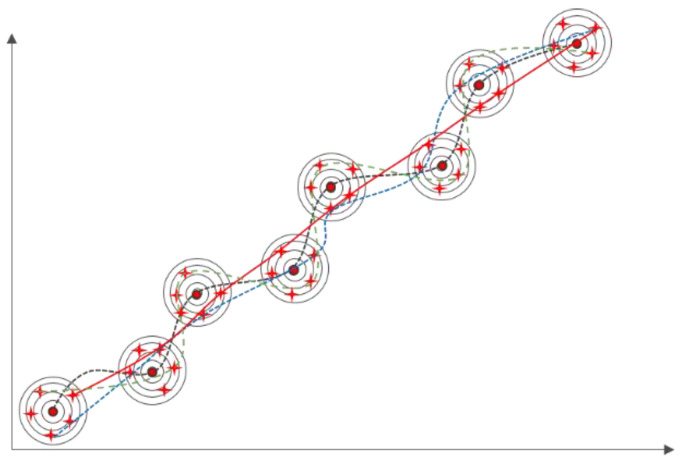
Schematic of the alternative trajectories: The colored lines are alternate trajectories, the red stars are particles, the red line is actual trajectory.

**Figure 6 sensors-23-09120-f006:**
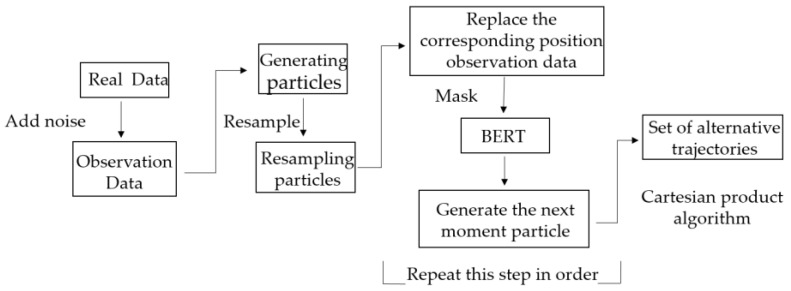
Flowchart of the particle filter algorithm to construct the set of alternative trajectories.

**Figure 7 sensors-23-09120-f007:**
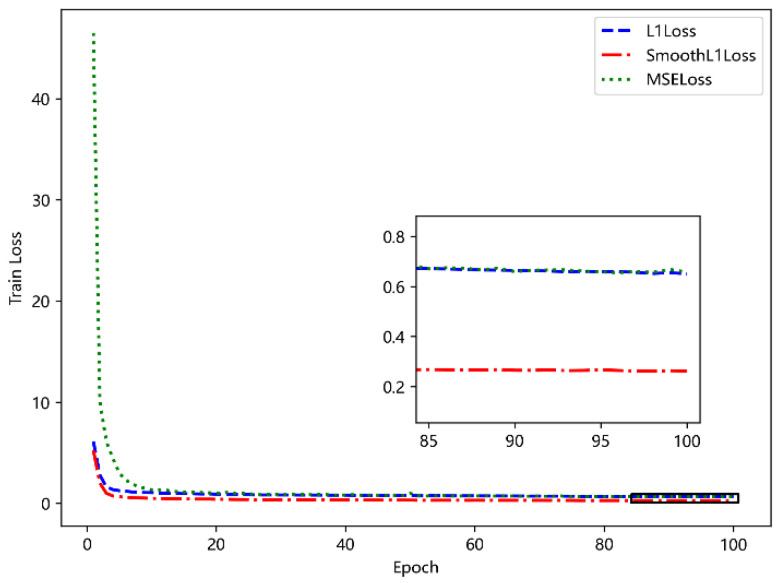
Comparison of different loss functions.

**Figure 8 sensors-23-09120-f008:**
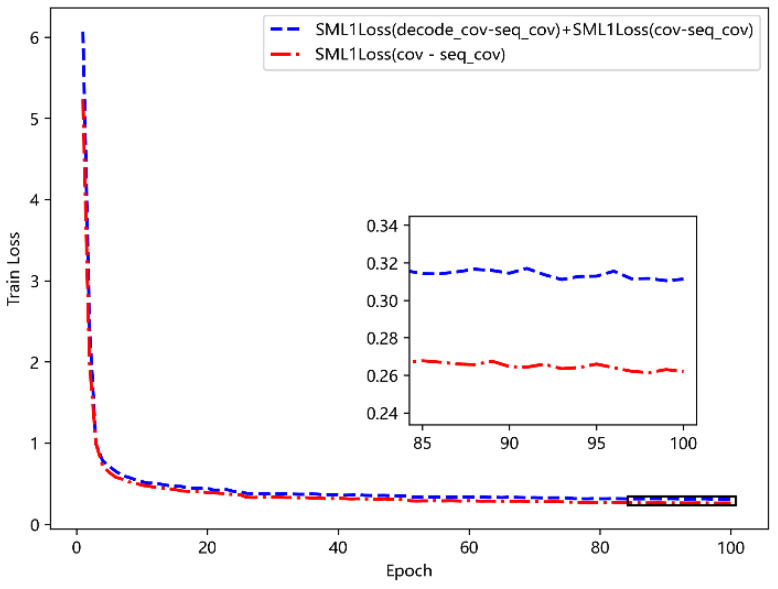
Comparison of different covariance loss functions.

**Figure 9 sensors-23-09120-f009:**
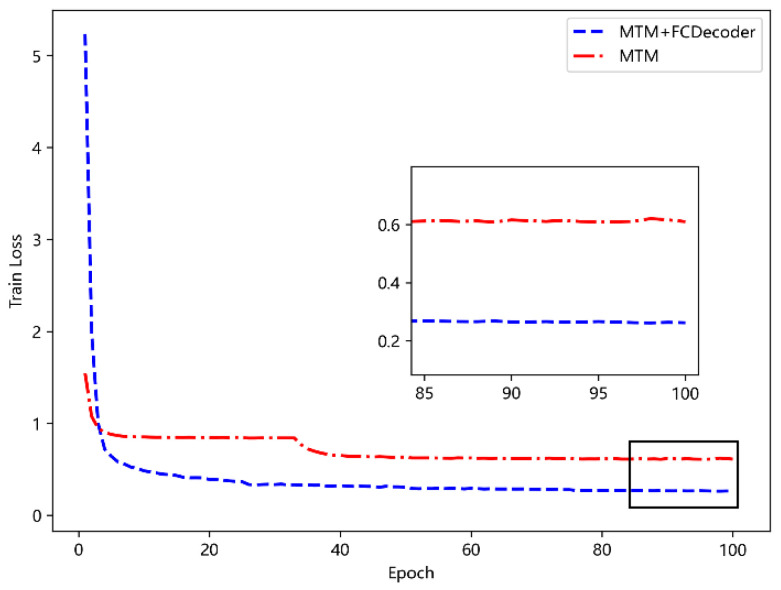
Compare the effect of FC-Decoder.

**Figure 10 sensors-23-09120-f010:**
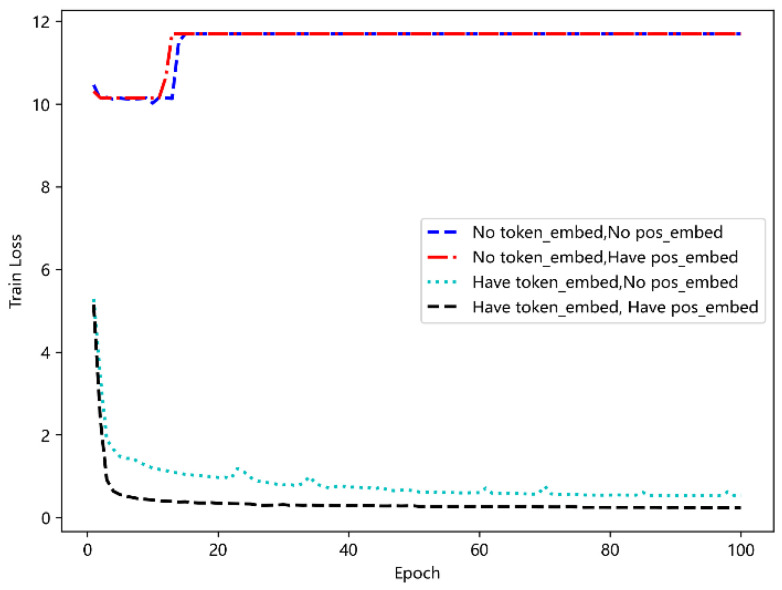
Comparing the effect of positional embedding and token embedding.

**Figure 11 sensors-23-09120-f011:**
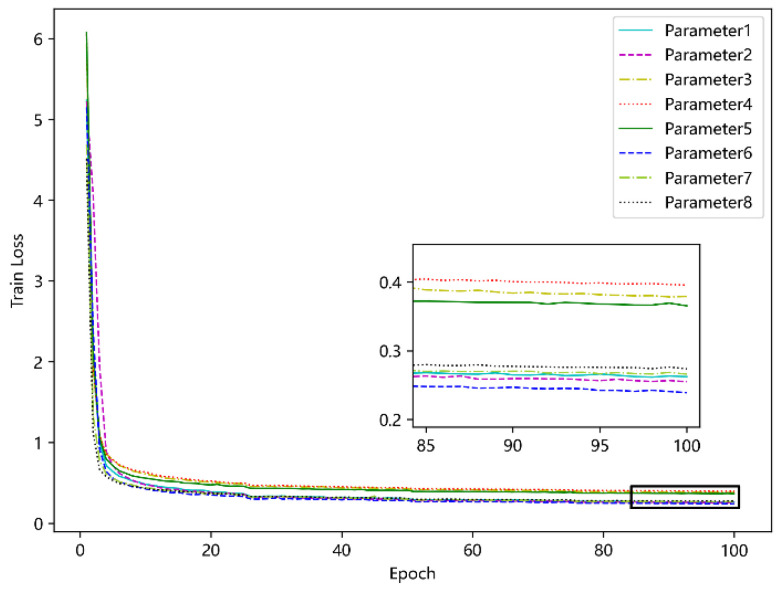
Comparison of different experimental parameters.

**Figure 12 sensors-23-09120-f012:**
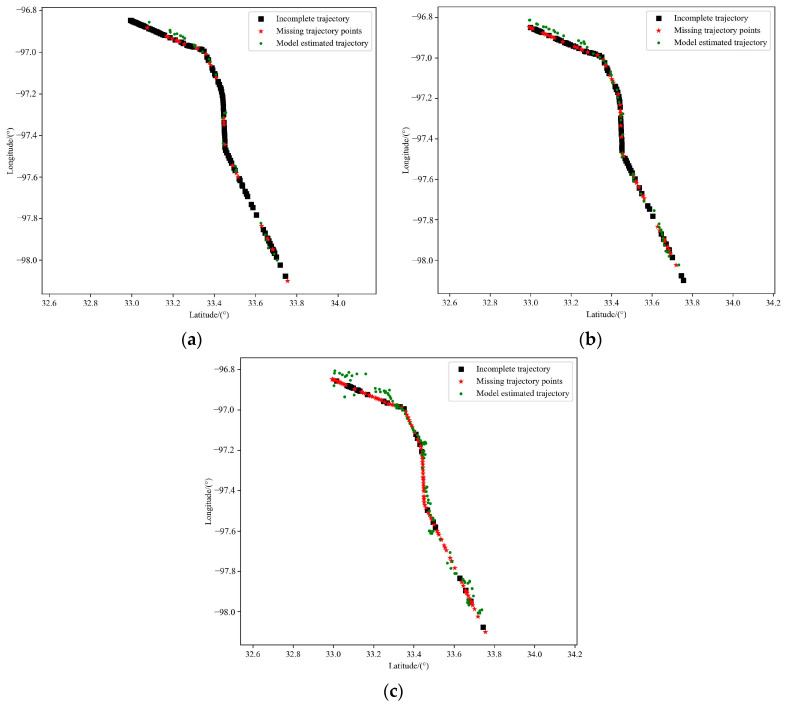
Verifying the model training effect: (**a**) Missing 20% of trajectory points; (**b**) Missing 40% of trajectory points; (**c**) Missing 80% of trajectory points.

**Figure 13 sensors-23-09120-f013:**
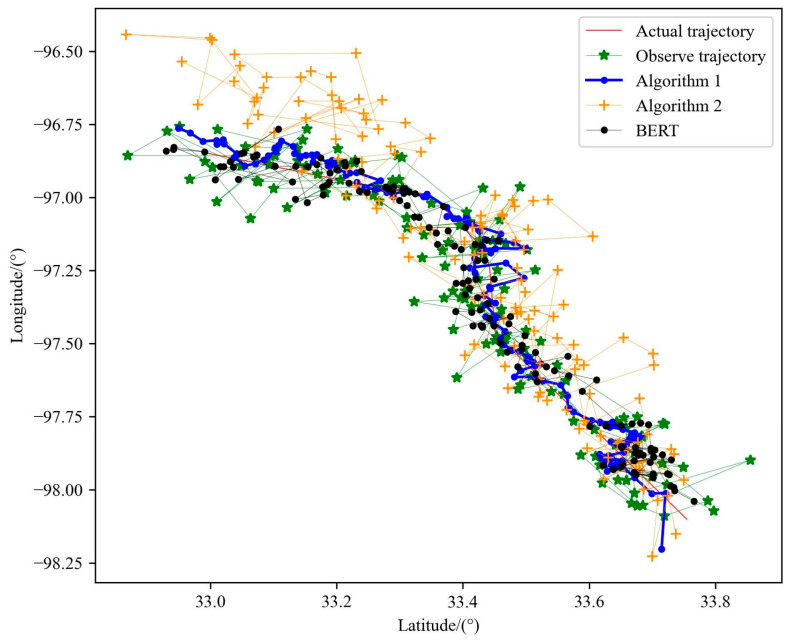
Performance comparison of two algorithms and BERT.

**Figure 14 sensors-23-09120-f014:**
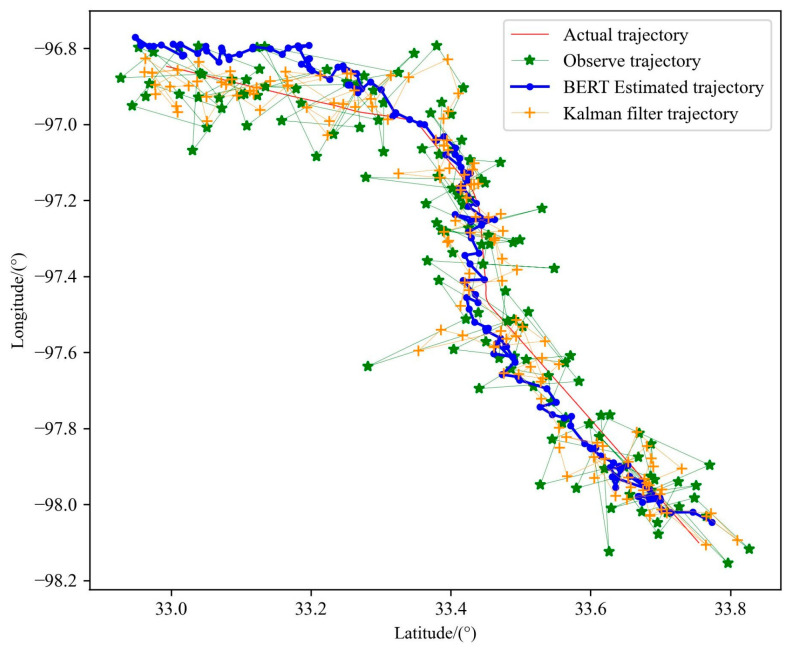
Performance comparison between BERT trajectory pre-training model and Kalman filter algorithm in trajectory 1.

**Figure 15 sensors-23-09120-f015:**
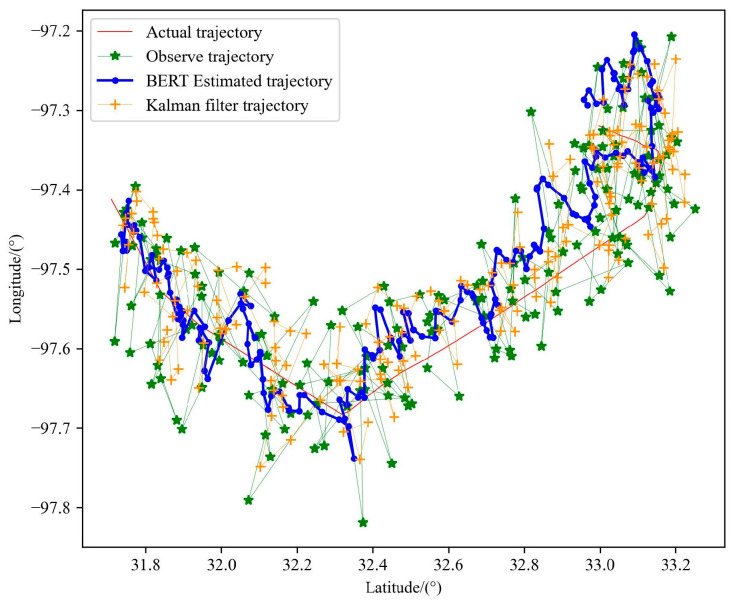
Performance comparison between BERT trajectory pre-training model and Kalman filter algorithm in trajectory 2.

**Table 1 sensors-23-09120-t001:** Experimental parameter.

Parameter	The Numerical
Optimizer	Adam
Rate of learning	l^−3^
Weight decline	l^−5^
Epoch	100
Batch Size	1024
Dropout	0.1
Sequence length	100

**Table 2 sensors-23-09120-t002:** Parameter Setting.

	Parameter Setting				Loss
	Input Dimensions	Output Dimensions	Number of Heads	Number of Encoder	
Parameter 1	128	256	4	4	0.2619
Parameter 2	256	256	4	4	0.2547
Parameter 3	128	128	4	4	0.3783
Parameter 4	64	128	4	4	0.3956
Parameter 5	256	128	4	4	0.3653
Parameter 6	256	256	2	4	0.2387
Parameter 7	256	256	4	2	0.2657
Parameter 8	256	256	2	2	0.2734

**Table 3 sensors-23-09120-t003:** Trajectory evaluation metrics for different proportions of missing points.

Percentage of Missing Points	Evaluation Indicators	
	MAE	MEDV/km
20%	0.00319	0.484
40%	0.00873	1.358
80%	0.02123	3.391

**Table 4 sensors-23-09120-t004:** Comparison of the evaluation indicators of the two algorithms and BERT.

Parameter		Trajectory Data	MAE	MEDV/km
Number of Particles	Number of Alternative Trajectories			
\	\	Observe Trajectory	0.05115	8.183
15	500,000	Algorithm A Trajectory	0.02843	4.492
15	500,000	Algorithm B Trajectory	0.09796	15.408
\	\	BERT	0.03015	4.860

**Table 5 sensors-23-09120-t005:** Comparison of the evaluation indicators of BERT trajectory pre-training model and Kalman filter algorithm in trajectory 1.

Parameter		Trajectory Data	MAE	MEDV/km
Number of Particles	Number of Alternative Trajectories			
\	\	Observe Trajectory	0.05342	8.301
15	500,000	Trajectory estimation by BERT	0.03104	4.913
\	\	Trajectory estimation by Kalman filter	0.03522	5.529

**Table 6 sensors-23-09120-t006:** Comparison of the evaluation indicators of BERT trajectory pre-training model and Kalman filter algorithm in trajectory 2.

Parameter		Trajectory Data	MAE	MEDV/km
Number of Particles	Number of Alternative Trajectories			
\	\	Observe Trajectory	0.04665	7.390
15	500,000	Trajectory estimation by BERT	0.03508	5.435
\	\	Trajectory estimation by Kalman filter	0.03887	6.267

## Data Availability

Data are contained within the article.

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
