# Peer review of "Trajectory-BERT: Trajectory Estimation Based on BERT Trajectory Pre-Training Model and Particle Filter Algorithm"

_sensors, 2023, doi:10.3390/s23229120_

Round 1

Reviewer 1 Report

Comments and Suggestions for Authors

This paper proposes a trajectory estimation method based on the BERT (Bidirectional Encoder Representations from Transformers) trajectory pre-training model combined with a particle filter algorithm (among two algorithms studied and compared), to solve the problem by masking ADS-B (Automatic Dependent Survey Broadcast) trajectory data with high accuracy and estimating the mask position value based on forward and backward trajectory points during BERT model training, using (Adam optimizer). The authors used data sourced from the ADS-B public dataset (between January and May 2021) provided by the OpenSky website, which meets (according to the paper) the data accuracy requirements of the model.

In the results section, the authors presented several comparisons such as comparison of different loss functions, comparison of different covariance loss functions. They also studied the effect of the FC-Decoder layer on BERT model training, which helped determine optimal model parameters. At the end, they presented a Performance Comparison between the BERT trajectory pre-training model and the Kalman filter algorithm.

- The method presented has a number of advantages. Generally speaking, the article is well presented and structured, and the theoretical part is well explained.

- Authors should improve the level of English in the manuscript.

- Some figures and titles should start with words, not verbs.

- Figure labels are unclear, and the font size should be larger.

Comments on the Quality of English Language

Authors should improve the level of English in the manuscript.

Reviewer 2 Report

Comments and Suggestions for Authors

The authors present a novel method for aircraft trajectory estimation using a BERT pre-training model and particle filter algorithm in cases with noise or missing data points. The main contributions are pre-training BERT on continuous trajectory sequences, combining this with a particle filter to generate candidate trajectories, and selecting the optimal trajectory by maximum a posteriori probability. The proposed methodology sounds fine, but I have some concerns about the novelty and evaluation of the method.

Major points:

1. The introduction would benefit from more clearly situating this work in the context of prior research on trajectory modeling. How does masking trajectories for BERT pre-training compare to previous techniques like trajectory vectorization or grid encodings?

2. The authors claim this approach does not require manual trajectory modeling, but the particle filter still relies on assumptions about motion dynamics. More comparison on the benefits of learning representations versus model-based methods would strengthen the motivation.

3. Additional ablation studies could demonstrate the impact of the different components, like examining performance without particle filter candidates or without the FC-Decoder branch.

4. For a more robust evaluation, testing on trajectories from other datasets or geographic regions besides the single aircraft would better showcase the generalization capability.

Minor points:

1. The writing could be improved in certain areas for clarity, such as expanding acronyms upon first use.

2. Consider adding pseudocode for the overall pipeline to improve readability.

3. Grammar check.

Comments on the Quality of English Language

Moderate editing of English language required

Reviewer 3 Report

Comments and Suggestions for Authors

Comments on the Quality of English Language

1) I recommend using more formal and shorter constructions. For example:

In line 144, please replace "Unlike" with "In contrast with";

In line 171, please replace "So that" with "Thus,"

In line 172, please replace "At the same time" with "Simultaneously"

2) There are many long sentences in the manuscript that can make it difficult to read (for example, the sentence from line 95). I recommend that you try to break some of them down into shorter sentences (one sentence have one idea).

Round 2

Reviewer 2 Report

Comments and Suggestions for Authors

The revision addressed my comments. Overall the paper is acceptable.